# Room-temperature terahertz anomalous Hall effect in Weyl antiferromagnet Mn$_3$Sn thin films

Takuya Matsuda[1], Natsuki Kanda[1], Tomoya Higo [1,2], N.P. Armitage[3], Satoru Nakatsuji [1,2,3,4] & Ryusuke Matsunaga [1,5 ✉]

Antiferromagnetic spin motion at terahertz (THz) frequencies attracts growing interests for fast spintronics, however, their smaller responses to external field inhibit device application. Recently the noncollinear antiferromagnet Mn$_3$Sn, a Weyl semimetal candidate, was reported to show large anomalous Hall effect (AHE) at room temperature comparable to ferromagnets. Dynamical aspect of such large responses is an important issue to be clarified for future THz data processing. Here the THz anomalous Hall conductivity in Mn$_3$Sn thin films is investigated by polarization-resolved spectroscopy. Large anomalous Hall conductivity Re $\sigma_{xy}(\omega) \sim 20\ \Omega^{-1}\mathrm{cm}^{-1}$ at THz frequencies is clearly observed as polarization rotation. A peculiar temperature dependence corresponding to the breaking/recovery of symmetry in the spin texture is also discussed. Observation of the THz AHE at room temperature demonstrates the ultrafast readout for the antiferromagnetic spintronics using Mn$_3$Sn, and will also open new avenue for studying nonequilibrium dynamics in Weyl antiferromagnets.

[1] The Institute for Solid State Physics, The University of Tokyo, Kashiwa, Chiba 277-8581, Japan. [2] CREST, Japan Science and Technology Agency, Kawaguchi, Saitama 332-0012, Japan. [3] The Institute of Quantum Matter, Department of Physics and Astronomy, The Johns Hopkins University, Baltimore, MA 21218, USA. [4] Department of Physics, The University of Tokyo, Hongo, Bunkyo-Ku, Tokyo 113-0033, Japan. [5] PRESTO, Japan Science and Technology Agency, 4-1-8 Honcho, Kawaguchi, Saitama 332-0012, Japan. ✉email: matsunaga@issp.u-tokyo.ac.jp

The control of magnetism has been a key issue for modern data processing and recording in spintronic devices. From the viewpoint of manipulation speed, antiferromagnets are promising materials since spin precession motion occurs typically at terahertz (THz) frequencies, a few orders of magnitude higher than ferromagnets[1]. Therefore, optical control of the antiferromagnets have been intensively investigated in the past few decades[2]. Nonlinear interaction with THz magnetic field has been also studied[3], which leads to direct spin manipulation for future data processing in nonvolatile memory devices. Readout of the magnetization information in antiferromagnets is, however, still very difficult since they are generally not sensitive to external field because of the much smaller net magnetization than in ferromagnets, which has made their practical application challenging. On the other hand, recent discovery of Weyl semimetals with inversion or time-reversal symmetry breaking have attracted tremendous interests[4] not only for fundamental physics as condensed matter analog of massless Weyl fermions but also for highly intriguing response functions that reflect the broken symmetry such as nonreciprocal or second-order nonlinear responses[5,6]. In particular, recent reports in the broken time-reversal symmetry Weyl semimetal candidate noncollinear antiferromagnet $Mn_3X$ ($X$ = Sn, Ge)[7–9] have opened new pathways for utilizing these materials as functional spin-based devices at room temperature. In spite of the vanishingly small net magnetization in the antiferromagnetic phase, these materials show large anomalous Hall effect (AHE)[7–9], anomalous Nernst effect[10,11], magneto-optical Kerr effect[12], and magnetic spin Hall effect[13] in a stark contrast to conventional antiferromagnets. These large responses comparable to ferromagnets have been attributed to the Berry curvature in momentum space[14–18], which is in particular enhanced at the Weyl nodes. Such Weyl semimetals, or Weyl (antiferro)magnets, are of great interest since the magnetic ordering can be controlled by temperature or external field, as small coercive field of 0.05 T has been reported for bulk $Mn_3Sn$[7]. A recent neutron scattering experiment has also revealed THz spin excitation in $Mn_3Sn$[19]. Therefore, deep understanding of the dynamical properties of $Mn_3Sn$ in the THz frequency range would contribute to developing novel devices based on fast motion of antiferromagnetic spins and large response to external field.

Previously, the anomalous Hall conductivity spectrum $\sigma_{xy}(\omega)$ at THz frequencies has been investigated in ferromagnets with the perspective to reveal the microscopic origin of the AHE[20–22]. In general, the AHE could arise from the intrinsic Berry curvature determined by the electronic band structure[23,24] or the extrinsic skew scattering or side jump with impurities[25,26]. If it is intrinsic, the AC anomalous Hall conductivity spectrum shows peculiar resonant structures due to interband transition as presented in infrared or THz spectroscopy[20–22]. The AHE in THz frequency has been also observed in a ferromagnetic semiconductor[27] or quantum anomalous Hall state in a topological insulator[28]. For Weyl semimetals, however, the THz anomalous Hall conductivity remains unexplored experimentally, although a number of theoretical efforts have been devoted to investigating the AC AHE across the Weyl nodes[29–33]. A recent angle-resolved photoemission spectroscopy (ARPES) for $Mn_3Sn$ has captured the Weyl-like dispersion at several meV above the Fermi energy $E_F$[34]. Therefore, low-energy THz polarimetry will provide deep insight into the microscopic picture of the large AHE in the Weyl semimetal.

In this work, using $Mn_3Sn$ thin films showing the large AHE comparable to the bulk[35], we perform THz time-domain spectroscopy (THz-TDS) with precise polarization resolution to reveal the THz anomalous Hall conductivity. Polarization rotation is clearly observed at room temperature and zero magnetic field, which quantitatively agrees with the large AHE previously

reported in the DC resistivity measurement. The results also demonstrate small dissipation in the AHE up to THz frequencies, which is consistent with an intrinsic origin of it. The temperature dependence of the Hall conductivity is also discussed with regards to macroscopic time-reversal symmetry in the spin order.

## Results

**Sample.** Samples used in this work are polycrystalline $Mn_{3+x}Sn_{1-x}$ thin films ($x$ = 0.00, 0.02, and 0.08) deposited on $SiO_2$ or thermally oxidized Si substrates by DC magnetron sputtering[35]. Figure 1a, b shows a 3D schematic view of atomic configuration and the top view along $c$-axis of magnetic structure, respectively. Figure 1c presents the X-ray diffraction (XRD) patterns of the 50-nm-thick film of $Mn_{3+x}Sn_{1-x}$ ($x$ = 0.02) on the $SiO_2$ substrate obtained by the grazing angle XRD measurement. All the peaks can be indexed by the hexagonal $D0_{19}$ $Mn_3Sn$ structure and no additional peaks coming from plausible impurity phases are observed, which is consistent with that in the films on a thermally oxidized Si substrate[35].

Figure 1d provides a schematic diagram of our sample configuration. Below the Néel temperature 420 K, the spins on Mn atoms order in an inverse triangular spin structure, where the spins form a 120° order with negative vector chirality in the $ab$ plane. Such a noncollinear antiferromagnetic spin in the Kagome bilayer is characterized by cluster magnetic octupole moments[18], which macroscopically breaks time-reversal symmetry and is expected to result in the appearance of Weyl nodes[14–18]. The small net magnetization occurs in the $ab$ plane by slight canting of the 120° order. Note that the large AHE is not explained by the small net magnetization but rather attributed to the cluster magnetic octupole[12]. An external magnetic field is applied normal to the film surface to align the cluster magnetic octupole along $z$-direction as well as the small magnetization vector. Thus, the electric field parallel to the sample surface ($x$-direction) induces the anomalous Hall conductivity $J_y = \sigma_{xy}E_x^{in}$ as well as the longitudinal conductivity $J_x = \sigma_{xx}E_x^{in}$, which are measured in this work in THz frequency.

**THz spectroscopy.** By conventional transmission THz-TDS (see Methods), THz longitudinal conductivity spectra $\sigma_{xx}(\omega)$ from 2 to 10 meV ($\omega/2\pi$ = 0.5 to 2.5 THz) were obtained in the thin-film approximation. Figure 1e, f shows real and imaginary parts of $\sigma_{xx}(\omega)$ in $Mn_{3+x}Sn_{1-x}$ thin films at various temperatures ($x$ = 0.02). We fit the data by using the Drude model $\sigma_{xx}(\omega) = \sigma_0/(1 - i\omega\tau)$, where $\sigma_0$ is DC conductivity and $\tau$ is carrier scattering time. The solid curves in Fig. 1e, f represent the result of fitting. $\tau$ as a function of temperature is shown in the inset to Fig. 1f. The scattering time is sensitive to temperature and becomes short with increasing temperature as is usual in metals. Above 250 K, however, the temperature dependence weakens and almost saturates, which well reproduces the previous THz-TDS measurement for a $Mn_3Sn$ thin film[36] and is discussed later.

Note that, although the THz longitudinal conductivity $\sigma_{xx}(\omega)$ in Fig. 1e coincides with the DC value because of the short scattering time, it does not mean that the THz anomalous Hall conductivity $\sigma_{xy}(\omega)$ is also in the DC limit. This is because the microscopic mechanisms are different between $\sigma_{xx}(\omega)$ and $\sigma_{xy}(\omega)$. $\sigma_{xx}(\omega)$ is well described with Drude model where any kinds of momentum scattering are involved. On the other hand, $\sigma_{xy}$ occur without any scattering process or only with impurity-induced scattering, depending on its intrinsic or extrinsic origin. In particular, in addition to the Weyl semimetallic bands, $Mn_3Sn$ also has other metallic bands[34] which dominates the longitudinal conductivity. Since the intrinsic AHE origantes from the integrated Berry curvature of the occupied states over the entire

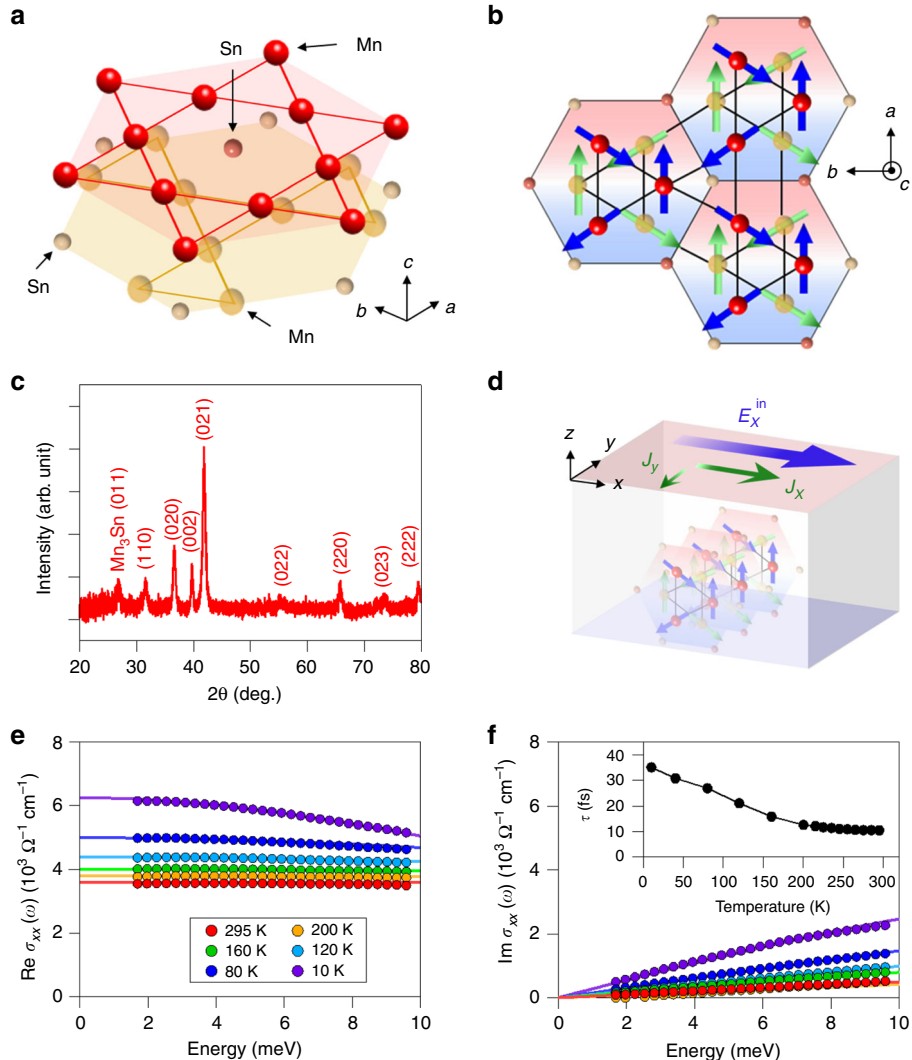

**Fig. 1 Crystal and magnetic structures of Mn₃Sn and the THz longetudinal conductivity spectra. a, b** A 3D schematic view of atomic configuration (**a**) and top view along c-axis (**b**) of the magnetic structures of Mn₃Sn at room temperature where the magnetic moments form an inverse triangular spin structure in the ab plane. **c** The X-ray diffraction measurement for the 50-nm-thick film of $Mn_{3+x}Sn_{1-x}$ ($x = 0.02$) on a SiO₂ substrate. **d** A schematic of our sample configuration. $E_x^{in}$ the incident electric field polarized along the x direction, $J_x$ the longitudinal current, $J_y$ the Hall current. **e, f** The real and imaginary parts of THz longitudinal conductivity spectra of $Mn_{3+x}Sn_{1-x}$ thin films ($x = 0.02$) on a SiO₂ substrate at various temperatures. The solid curves are the results of the Drude-model fitting. The inset shows temperature dependence of the scattering time obtained from the fitting.

Brillouin zone, the frequency dependence of $\sigma_{xy}$ is not obvious even if the scattering time in $\sigma_{xx}$ is known and therefore to be investigated experimentally.

Figure 2a schematically shows our polarization-resolved THz-TDS setup with a set of freestanding wire-grid polarizers (WGP)[37–40]. WGP1 defines the polarization of the incident THz electric field as x direction, and WGP3 is set to block the x-component THz field before detection. By using two types of configurations for WGP2 (configs. 1 and 2 in the inset of Fig. 2a), both $E_x(\omega)$ and $E_y(\omega)$ are obtained. In the small-angle limit, the rotation-angle and ellipticity-angle spectra, $\theta(\omega)$ and $\eta(\omega)$, are expressed as $\theta(\omega) + i\eta(\omega) \sim E_y(\omega)/E_x(\omega)$. Here, WGP3 and the ZnTe crystal are set as the detection efficiency of the y-component electric field is maximized[41] (see details in Methods). To evaluate the precision of our measurement, without samples we performed 1 scan and 10 scans of THz-TDS with the configs. 1 and 2, respectively, which gives one data set of the $\theta(\omega)$ and $\eta(\omega)$ spectra within 60 s. With increasing the number of the data sets, the standard deviation of $\theta(\omega)$ is plotted as a function of frequency in

Fig. 2b. Even for one data set, the standard deviation is smaller than 0.5 mrad in the frequency range from 0.5 to 2.0 THz (2 to 8 meV). The precision can be further improved as small as several tens of μrad between 0.5 and 2.0 THz with 20-min accumulation time. This result ensures high-precision spectroscopy of polarization rotation in this frequency window.

**Large THz AHE at room temperature**. The polarization-resolved THz-TDS measurements for $Mn_{3+x}Sn_{1-x}$ thin films were performed at zero magnetic field after the samples are magnetized under field of 5 T, where the magnetization vector **M** is normal to the film surface. Figure 2c, d shows the $\theta(\omega)$ and $\eta(\omega)$ spectra, respectively, with different film thicknesses of $d = 50$, 200, and 400 nm at room temperature ($x = 0.02$). With using a bare SiO₂ substrate as a reference, the polarization rotation of ~4 mrad is observed for a 50-nm sample, and the rotation angle increases as the film thickness increases. The broken curves in Fig. 2c, d show the data taken upon flipping the sample to get an oppositely directed magnetization vector (−**M**), which

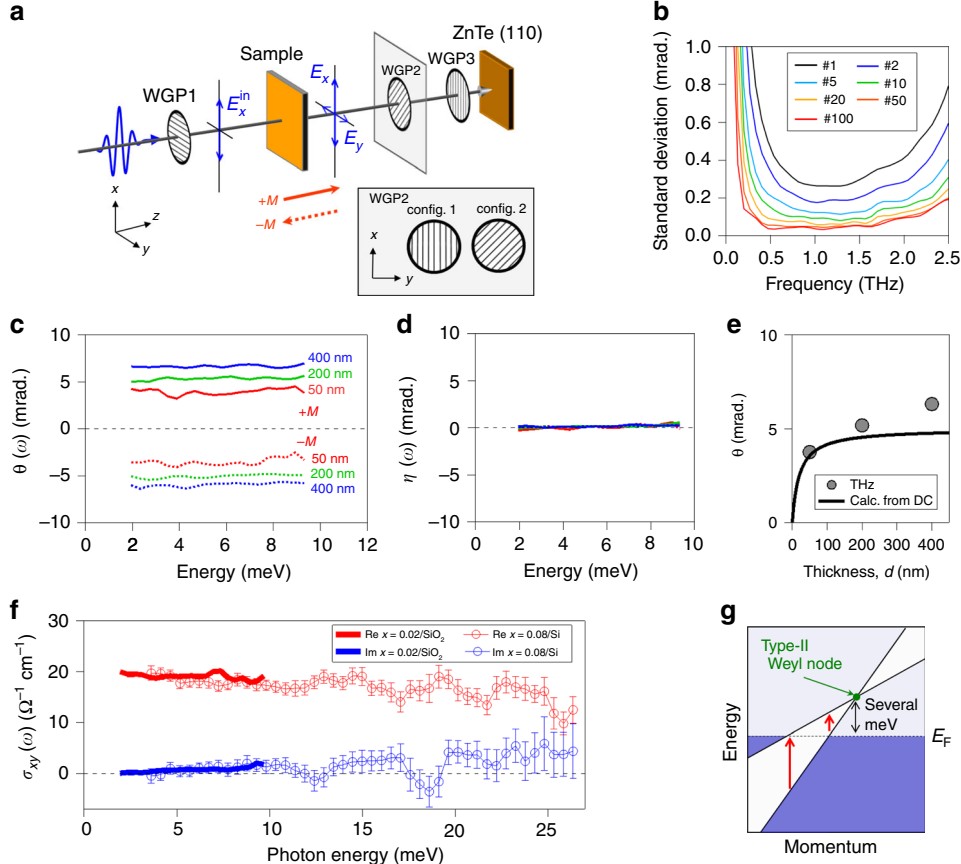

**Fig. 2 THz anomalous Hall effect at room temperature with polarization-resolved spectroscopy. a** A schematic of our polarization-resolved measurement setup. WGP wire-grid polarizer. **b** Frequency dependence of the precision of the polarization rotation angle in this measurement evaluated by the standard deviation of the polarization rotation angle. The precision can be more improved with using a larger number (#) of data sets. For example, the precision for 20 data sets (#20) can be as small as several tens of μrad between 0.5 and 2.0 THz (See details in text and Methods). **c, d** The rotation-angle and ellipticity-angle spectra in $Mn_{3+x}Sn_{1-x}$ films ($x = 0.02$) with different film thicknesses at room temperature. The broken curves correspond to the data with a flipped sample for the opposite magnetization vector. **e** Filled circles are the averaged rotation angle as a function of the film thickness. The solid line is the calculation of Eq. (1) fixed the DC longitudinal and Hall conductivity for 200-nm-thick sample. **f** The real- and imaginary-part Hall conductivity spectra for $Mn_{3+x}Sn_{1-x}$ films. The solid curves show the low-frequency THz-TDS for $x = 0.02$ on a $SiO_2$ substrate and the open circles show the broadband spectrum for $x = 0.08$ on a Si substrate. **g** A schematic of interband transition across the type-II Weyl nodes. The error bars in **f** indicate the standard deviations for the statistical fluctuation after repeating the measurements.

corresponds to the reverse of the cluster octupole moment. Upon flipping, the signs of the rotation angle are reversed. The filled circles in Fig. 2e show the thickness dependence of $\theta(\omega)$ averaged over this energy range calculated by the following form

$$\theta = \frac{\sigma_{xy} Z_0 d}{1 + n_s + \sigma_{xx} Z_0 d},\tag{1}$$

where the vacuum impedance $Z_0 = 377\,\Omega$ and the substrate refractive index $n_s = 1.92$. With the longitudinal conductivity $\sigma_{xx} = 3800\,\Omega^{-1}\mathrm{cm}^{-1}$ and the Hall conductivity $\sigma_{xy} = 19\,\Omega^{-1}\mathrm{cm}^{-1}$, which was evaluated for a 200-nm-thick film in DC resistivity measurements, the calculation reasonably reproduces our THz experimental data (solid line). In the small thickness regime ($d \ll 50\,\mathrm{nm}$), $\theta \sim \sigma_{xy} Z_0 d/(1 + n_s)$, and therefore $\theta$ is proportional to the film thickness $d$. In the large thickness regime ($d \gg 50\,\mathrm{nm}$), $\theta$ is saturated to be $\sim \sigma_{xy}/\sigma_{xx}$. Slight difference of the magnitudes in $\theta$ could be attributed to the difference of geometry; the DC resistivity is measured with electrodes attached on the film surface while the THz conductivity is measured in transmission. These results clearly indicate that the large AHE in $Mn_3Sn$ also appears in the THz frequency range as a polarization rotation. It may be also noteworthy that the DC AHE is usually measured under magnetic field

by changing its sign to avoid creation of magnetic domains under zero field and to detect difference of Hall resistivity between positive and negative field with removing artifact such as contact resistance. In the present experiment, the Hall conductivity can be clearly observed as light polarization rotation in a noncontact fashion without using magnetic field. We also confirmed that the polarization rotation was unchanged even several months after the sample fabrication and magnetization, exhibiting robustness of magnetic information in the $Mn_3Sn$ thin films[35].

Figure 2d shows that $\eta(\omega)$ is negligibly small for all the samples at room temperature. It is Kramers–Kronig relation consistent with the fact that $\theta(\omega)$ is flat with a value expected from DC. Using Eq. (1), we obtained the real and imaginary parts of the THz anomalous Hall conductivity spectra $\sigma_{xy}(\omega)$ and plotted in Fig. 2f as solid curves. Im $\sigma_{xy}(\omega)$ at $\omega/2\pi = 1$ THz (~4 meV) is as small as ~0.4 $\Omega^{-1}\mathrm{cm}^{-1}$ and become much smaller for lower frequency in contrast to the large anomalous Hall conductivity Re $\sigma_{xy}(\omega) \sim 20\,\Omega^{-1}\mathrm{cm}^{-1}$ at the THz frequency. Since Im $\sigma_{xy}(\omega)$ indicates the dissipative part of the Hall current[24], this result provides a direct evidence for dissipationless feature of the anomalous Hall current in $Mn_3Sn$ up to the THz frequency scale, which is also consistent with the intrinsic nature of the AHE

driven by the large Berry curvature in momentum space. A recent study of thermal transport in $Mn_3Sn$ has also reported negligible inelastic scattering with phonons in the AHE[42].

To investigate in broader frequency range, we also performed the same polarization-resolved THz-TDS for a 50-nm $Mn_{3+x}$ $Sn_{1-x}$ ($x = 0.08$) on a Si substrate at room temperature by using (110) GaP crystals and 20-fs laser pulses with 1-kHz repetition (see Methods). The precision of our broadband measurement was evaluated in the same way as Fig. 2a (see Methods). The open circles in Fig. 2f shows $\sigma_{xy}(\omega)$ spectra up to ~7 THz, in which data with a flipped sample was used as a reference. The broadband data are smoothly connected to the low-frequency measurement. The gradual growths of the dissipative-part Hall conductivity Im $\sigma_{xy}(\omega)$ with increasing photon energy can be attributed to a broad optical transition across the band-crossing (or anticrossing) points as discussed for ferromagnets[20,22]. While the onset of the interband transition coincides with twice the chemical potential for the model proposed in a ferromagnet[22], Im $\sigma_{xy}(\omega)$ in Weyl semimetals is expected to show a different spectrum associated with anisotropic cone dispersion[31,32] which is inherent to a pair of Weyl nodes. A recent ARPES study for $Mn_{3+x}Sn_{1-x}$ ($x = 0.03$) has revealed the Weyl-like dispersions at ~8 meV above $E_F$ with strong band renormalization by a factor of 5 in comparison with the first-principle calculation[34]. Notably these Weyl nodes are type-II[17,34], where both electron and hole pockets exist with a strongly anisotropic dispersion as illustrated in Fig. 2g. The asymmetric dispersion indicates that a broad absorption could occur even below 8 meV[32]. In the present experiment, onset of the interband transition was not clearly discerned, perhaps because it could be obscured by the room temperature thermal energy.

**Temperature dependence of THz AHE**. We also measured the temperature dependence of the THz AHE for $Mn_{3+x}Sn_{1-x}$ films ($x = 0.02$) on a $SiO_2$ substrate. The sample was magnetized at 300 K in advance and cooled down under zero magnetic field. Figure 3a–d shows Re $\sigma_{xy}(\omega)$ and Im $\sigma_{xy}(\omega)$ at higher temperature (300–200 K) and lower temperatures (160–10 K). Figure 3e exhibits Re $\sigma_{xy}(\omega)$ averaged between 2 and 10 meV as a function of temperature. As the temperature decreases, Re $\sigma_{xy}(\omega)$ is sharply suppressed around 250 K. The similar sharp reduction of the AHE as well as magnetization has been also observed in the DC measurement[35]. Such a spin-reorientation phase transition has been studied by neutron scattering experiments, which have revealed that the 120° spin order in each $ab$ plane rotates to form a helical spin ordering along the $c$-axis below 250 K[43]. Interestingly, although the inverse triangular spin structure in the $ab$ plane of the Kagome bilayer below 420 K breaks macroscopic time-reversal ($T$) symmetry, the helical spin ordering that develops along the $c$-axis below 250 K recovers the macroscopic $T$-symmetry again, which results in the disappearance of the net Berry curvature. Therefore, the drastic reduction of THz AHE at low temperature as well as DC AHE is ascribed to the spin-reorientation phase transition to the helical structure[44–46]. In addition to the polarization rotation, a peculiar temperature dependence is also found in the scattering time $\tau$ obtained from $\sigma_{xx}(\omega)$ in the inset of Fig. 1f. The scattering time decreases with increasing temperature, but shows saturation at around 250 K, which agrees well with the recent report[36]. This result might be also related to the appearance of the Weyl nodes around which backscattering could be suppressed.

Figure 3e shows that Re $\sigma_{xy}(\omega)$ increases slightly below 150 K. The Re $\sigma_{xy}(\omega)$ spectra at the low temperatures in Fig. 3c shows a slope upward toward lower frequency, which is clearly distinct from the flat spectra at higher temperatures in Fig. 3a.

Correspondingly, Im $\sigma_{xy}(\omega)$ spectra in Fig. 3b, d also show qualitatively different behaviors. Previous studies have reported the emergence of the spin glass state at the low temperature with a weak ferromagnetism due to spin canting towards the $c$-axis[47,48]. Note that our THz measurement was performed at zero magnetic field on cooling after the demagnetization process (we first applied a field of 5 T perpendicular to the film surface and decrease the field down to 0 T) at 300 K. In such a situation the macroscopic magnetic moment is much smaller than that in case of field cooling due to random orientation of ferromagnetic domains. The peculiar Hall conductivity spectra at low temperatures in Fig. 3c, d would reflect the spatial inhomogeneity and might be described by the effective medium theory. Nevertheless, substantial dissipative part comparable to the real part at the low temperatures in Fig. 3d is in contrast to the room temperature Weyl semimetal phase, which implies that different mechanism including substantial dissipation might be involved with the AHE at the spin glass phase.

In summary, by using the polarization-resolved THz spectroscopy for the $Mn_3Sn$ thin films, we observed the large anomalous Hall conductivity of $Mn_3Sn$ in THz frequency with negligibly-small dissipation at room temperature. Such a large THz response in the antiferromagnets is desirable and paves the way for ultrafast readout of spin with THz current on spintronic device. The far-field THz polarimtry presented in this work is possible only for large area due to the diffraction limit. For more practical application, THz anomalous Hall conductivity must be measured in much smaller regions. Recently the light-induced THz AHE of a tiny graphene flake has been detected via contacted electrodes combined with optical pulses and THz current generation on chip[49]. Our demonstration of the THz AHE will lead to such a readout of the spin information on integrated devices.

Importantly, our all-optical approch in the noncontact way with picosecond time resolution can be extended to pump–probe measurements for the study of nonequilibrium dynamics. THz control of the cluster magnetic octupole in $Mn_3Sn$ is highly demanded for ultrafast writing. If the external strong magnetic field is applied to the in-plane opposite direction to the ground state, all of the spins on the Kagome bilayer change their directions simultaneously, resulting in the flip of the cluster magnetic octupole. In terms of the spin precession, it corresponds to the damping of the acoustic collective mode, which could occur at picosecond timescale due to the exchange interaction in the antiferromagnetic metal.

From fundamental point of view, optical control of Weyl antiferromagnetis is also highly intriguing as recently the THz field control of Weyl nodes has been demonstrated in a noncentrosymmetric Weyl semimetal[50]. Even in static regime, further investigation of THz responses with suppressing thermal energy in another noncollinear antiferromagnetic compound $Mn_3Ge$, where the inverse-triangular spin ordering survives at low temperature[8,9], will clarify interband transition around the Weyl nodes. Higher-frequency infrared polarimetry would be also important for direct comparison with first principle calculation to identify the energies of the multiple Weyl nodes from the Fermi surface.

## Methods

**Sample preparation and characterization**. $Mn_3Sn$ polycrystalline films (50–400 nm) were fabricated as reported in ref. [35] on thermally oxidized Si (Si/$SiO_2$) substrates and quartz ($SiO_2$) substrates by DC magnetron sputtering from a $Mn_{2.7}Sn$ target in a chamber with a base pressure of $<5 \times 10^{-7}$ Pa. The $Mn_3Sn$ layer was deposited at room temperature, and it was subsequently annealed at 500 °C for 1 h. The sputtering power and Ar gas pressure were 60 W and 0.4–0.6 Pa, respectively. The compositions of the $Mn_3Sn$ films were determined by scanning electron microscopy-energy dispersive X-ray spectrometry. A hexagonal $D0_{19}$ $Mn_3Sn$ phase was confirmed by XRD measurement in $Mn_{3+x}Sn_{1-x}$ thin films

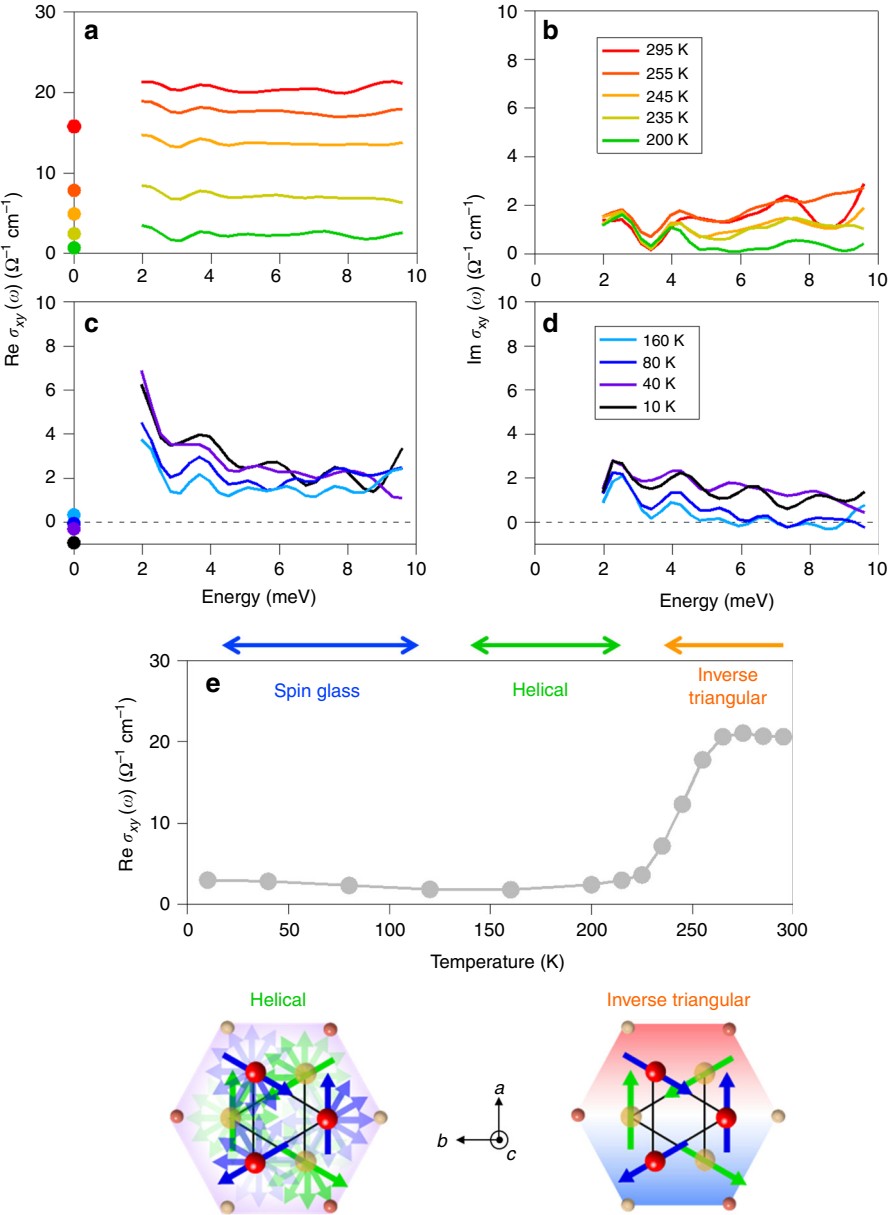

**Fig. 3 Temperature dependence of THz anomalous Hall conductivity spectra. a–d** The real and imaginary parts of the Hall conductivity for a sample ($x =$ 0.02) from 300 to 200 K (**a**, **b**) and from 160 to 10 K (**c**, **d**). The filled circles are the DC Hall conductivities at each temperature. **e** Temperature dependence of the real-part THz Hall conductivity. The lower panel shows the top views along $c$-axis of the magnetic structure at each phase.

($x = 0.00$, 0.02, and 0.08). The data for 50-nm-thick $Mn_{3+x}Sn_{1-x}$ film ($x = 0.02$) on a SiO$_2$ substrate is shown in Fig. 1c. These samples show the large AHE at room temperature comparable to the bulk single crystal reported[7], ensuring quality of the thin film.

**THz time-domain spectroscopy**. As a light source for THz-TDS, we used a mode-locked Ti:Sapphire laser with 800-nm central wavelength, 100-fs pulse duration, and the 76-MHz repetition rate (TSUNAMI, Spectra Physics). THz pulses were generated from an interdigitated photoconductive antenna on a GaAs substrate with 50-V biased voltage and 50-kHz modulation frequency. The transmitted THz pulses were detected by electro-optical (EO) sampling in a 2-mm-thick (110) ZnTe crystal with another sampling pulse. For broadband THz-TDS in Fig. 2f, we also used a regenerative amplified Ti:Sapphire laser system with 800-nm central wavelength, 50-fs pulse duration, and 1-kHz repetition rate (Spitfire Pro, Spectra Physics). Spectrum of the laser pulse is broadened by the self-phase modulation in SiO$_2$ plates and then the pulse width is compressed down to 20 fs by chirped mirrors. Broadband THz-TDS up to 7 THz (~28 meV) is realized by optical rectification and EO sampling with 300-μm-thick (110) GaP crystals.

In our experiment, we measured the sample and a bare substrate as a reference and then obtained the complex amplitude transmittance $t(\omega)$. In the thin-film approximation, the THz longitudinal conductivity spectrum Im $\sigma_{xx}(\omega)$ is obtained

from the relation

$$t(\omega) = \frac{1}{1 + n_{\mathrm{s}} + \sigma_{xx}(\omega)Z_0 d} \frac{4n_{\mathrm{s}}}{1 + n_{\mathrm{s}}} e^{i\Phi(\omega)}, \qquad (2)$$

where $Z_0$ is the vacuum impedance, $d$ is the film thickness, $n_s$ is the refractive index of the substrate, and $\Phi(\omega) = (n_{\mathrm{s}}d_{\mathrm{s}} - d - d_{\mathrm{s}})\omega/c$, where $d_s$ is the substrate thickness.

**Polarization-resolved THz TDS**. Schematic of our polarization-resolved spectroscopy setup is shown in Fig. 2a. Before the sample, the incident THz electric field polarization is determined as the $x$-direction by the first WGP1. The $x$- and $y$-components of the THz field just after transmitting the sample are defined as $E_x$ and $E_y$, respectively. After the sample, other two WGP (WGP2 and WGP3) are inserted before the EO sampling. The angle of WGP3 is fixed to block the $x$-component THz field, and the ZnTe crystal is set to maximize the detection efficiency of the $y$-component field. Such a sensitive detection of the tiny $y$-component THz field is important for high resolution of polarization[41]. The rotational angle of the WGP2 is controllable during the measurement. We define $\phi$ as the angle from the $x$-direction to the transmission direction of polarization for the WGP2. The

signal $F$ measured in the EO sampling is expressed as

$$F = \begin{bmatrix} 0 & 1 \end{bmatrix} R(\phi) T_{\mathrm{WGP2}} R(-\phi) \begin{bmatrix} E_x \\ E_y \end{bmatrix}, \qquad (3)$$

where $R(\phi)$ is the rotation matrix. $T_{\mathrm{WGP2}}$ is the complex Jones matrix of the WGP2, which is expressed as

$$T_{\mathrm{WGP2}} = \begin{bmatrix} t_{\parallel} & 0 \\ 0 & t_{\perp} \end{bmatrix}, \qquad (4)$$

where $t_{\parallel}$ and $t_{\perp}$ are the complex transmittances of the WGP2 for THz field polarization parallel and perpendicular to the wires. The Jones matrix $T_{\mathrm{WGP2}}$ is used to consider the effect of the finite extinction ratio of the WGP2 for accurate polarization measurement. As shown in the inset of Fig. 2a, we used two types of the WGP2 configurations, config. 1 ($\phi = 0°$) and config. 2 ($\phi = 45°$), and then we obtained the signals $F_1(t)$ and $F_2(t)$ and their Fourier components $F_1(\omega)$ and $F_2(\omega)$, respectively. By solving Eq. (3), $E_x(\omega)$ and $E_y(\omega)$ are obtained as

$$\begin{bmatrix} E_x(\omega) \\ E_y(\omega) \end{bmatrix} = \begin{bmatrix} 0 & t_{\perp}(\omega) \\ (t_{\perp}(\omega) - t_{\parallel}(\omega))/2 & (t_{\perp}(\omega) + t_{\parallel}(\omega))/2 \end{bmatrix}^{-1} \begin{bmatrix} F_1(\omega) \\ F_2(\omega) \end{bmatrix}, \qquad (5)$$

By using $E_x(\omega)$ and $E_y(\omega)$, the rotation-angle and ellipticity-angle spectra, $\theta(\omega)$ and $\eta(\omega)$, are expressed as

$$\theta(\omega) = \tan^{-1}\left( \frac{\mathrm{Re}\{E_x^*(\omega)E_y(\omega)\}}{|E_x(\omega)|^2 - |E_y(\omega)|^2} \right), \qquad (6)$$

$$\eta(\omega) = -\sin^{-1}\left( \frac{\mathrm{Im}\{E_x^*(\omega)E_y(\omega)\}}{|E_x(\omega)|^2 + |E_y(\omega)|^2} \right). \qquad (7)$$

In the small-angle limit, $\theta(\omega)$ and $\eta(\omega)$ are simply described as

$$\theta(\omega) + i\eta(\omega) \sim \frac{E_y(\omega)}{E_x(\omega)}. \qquad (8)$$

In the thin-film approximation, the anomalous Hall conductivity spectrum $\sigma_{xy}(\omega)$ is obtained from the relation

$$\sigma_{xy}(\omega) = \frac{\theta(\omega) + i\eta(\omega)}{Z_0 d}(1 + n_s + \sigma_{xx}(\omega)Z_0 d), \qquad (9)$$

which is reduced to Eq. (1) in the DC limit ($\omega \to 0$). The $\sigma_{xy}(\omega)$ spectra of the thin film at low temperature in Fig. 3a–d shows a small oscillation. It could be ascribed to interference of the THz pulses, which could be reflected from the cryostat window.

## Data availablity
The data that support the findings of this study are available from the corresponding authors upon reasonable request.

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

## Acknowledgements

The authors would like to thank Bing Cheng for access to his unpublished data, S. Miwa and K. Kuroda for fruitful discussion, Y. Kinoshita and M. Tokunaga for their help in applying magnetic field, and D. Nishio-Hamane for SEM-EDX measuremensts. This work was supported in part by JST PRESTO (Grant no. JPMJPR16PA), by JSPS KAKENHI (Grants Nos. 19H01817 and 19H00650), by JSPS Grants-in-Aid for Scientific Research on Innovative Areas "J-Physics" (Grants 15H05882 and 15H05883), and by JSPS Invitational Fellowships for Research in Japan. This work was partially support by JST CREST (Grants no. JPMJCR18T3). The work at JHU was supported through the Institute for Quantum Matter, an EFRC funded by the U.S. DOE, Office of BES under DE-SC0019331.

## Author contributions

R.M., S.N., and N.P.A. conceived this project. T.H. and S.N. performed the sample growth, characterization, and DC transport measurement. T.M., N.K., and R.M. developed the THz spectroscopy system, performed the experiments, and analyzed the data. All coauthors discussed the results. T.M. and R.M. wrote the paper with feedbacks from all the coauthors.

## Competing interests

The authors declare no competing interests.
