## [Peer Review File · Nature Communications]

Reviewers' comments:

Reviewer #1 (Remarks to the Author):

The authors present an experimental follow-up work on their earlier ground-breaking observation (Ref. 7) of the AHE in a non-collinear antiferromagnet Mn₃Sn. They now extend the AHE measurements from DC to the THz frequency range. The employed THz method and the presented data appear to be valid. In the present form of the manuscript, however, the authors did not evidence or explain in a transparent way a new unexpected phenomenology of the AHE in Mn₃Sn or help elucidating physical origins of the previously observed phenomenology. More specifically, I have the following comments:

1) Terminology: In their pioneering work from 2015 (Ref. 7), they call the DC AHE in single crystal Mn₃Sn a large effect. The observed DC AHE reached 30 Ohm⁻¹cm⁻¹ at room-T and 100 Ohm⁻¹cm⁻¹ at 100 K. In the present manuscript the values reach 20 Ohm⁻¹cm⁻¹. Also in comparison, in ferromagnetic Fe, AHE is around 1000 Ohm⁻¹cm⁻¹. The authors should therefore justify the use of the term “giant/gigantic” AHE in the present manuscript or use a more appropriate terminology.

2) The authors explore the frequency range of 2-27 meV which is well below their experimental value of the Drude scattering rate at room-T of 60meV ($\tau=10$ fs). So their AC experiment is deep in the Drude peak, i.e., still close to the DC limit. Under these conditions, the expected result is that the THz AHE would be similar to the DC AHE which at room-T indeed seems to be the case. The authors should note this in the manuscript or, if they feel that the observation was unexpected, they should explain it more explicitly.

3) As the authors mention in the paper, performing AHE measurements at THz is not novel on its own as this was done earlier in other magnetic systems. However, it is not a straightforward technique so the authors certainly do deserve a credit for their work here. On the other hand, because this is an elaborate technique, the reader might wonder how relevant are the remarks made by the authors on the relevance of their work for future high frequency device applications. The authors should elaborate in more detail on this potential relevance.

4) The seemingly surprising result is the drop of the THz AHE below 250K. The authors associate this observation with a transition to a helical state. However, they do not provide an additional experimental verification for this interpretation. Fig. 3c shows an upturn of the AHE at low frequencies which seems to extrapolate to a large value in the DC limit. This seems inconsistent with the scenario of the phase transition in to the helical state. It would be very helpful to see the AHE measured in these samples at lower temperatures as well. In Ref. 7, the DC AHE in single crystal Mn₃Sn does not show a drop at lower temperatures.

To conclude, it seems that among the presented data the ones at low temperatures might hint to some new interesting physics. However, the effect is not explored extensively and the tentative conclusions made by the authors seem not to be supported by the presented data.

Reviewer #2 (Remarks to the Author):

The authors report on very interesting work probing a terahertz anomalous hall response in Weyl antiferromagnetic systems. The all-optical approach is powerful here and as the authors state, opens up new possibilities for probing the non-equilibrium response. The clear evidence for dissipationless-like transport is also quite important. I only have a few minor suggestions which I think could improve the paper.

1. The authors write in the abstract "Observation of the THz AHE at room temperature demonstrates the ultrafast readout for the antiferromagnetic spintronics using Mn₃Sn, paving the path for ultrafast control of magnetism in Weyl antiferromagnets." My view is that the paper stands on its own without this last comment in the abstract which to me is a slight oversell. The approach does enable connection to pump-probe studies as noted above but does not address the means by which control of this system would be enabled. I think it's sufficient to describe this in the final discussion session as is currently done.

2. In ref 35 by the same authors, they see a similar drop in scattering time around 250K in the unmagnetized state. I would like to understand the relation here between these two measurements. Are these expected to give similar results?

3. I suggest that Figure 1e include more information in the figure caption itself rather than just referring the reader to the text. Right now this is not easily understandable.

4. The phrase "We also investigated the temperature dependence of the THz AHE for Mn_{3+x}Sn_{1-x} films (x=0.02) on a SiO₂ substrate at zero-field cooling after the magnetization at 300 K" needs rewording I think.

Overall I recommend publication of this work after the suggested minor changes. There will be significant interest in this in the broad community interested in topological and magnetic aspects of Weyl semimetals.

Reviewer #3 (Remarks to the Author):

This manuscript by T. Matsuda et al. reports polarization-resolved THz spectroscopy on a Weyl semimetal candidate Mn₃Sn, which is of current interest in the field of condensed matter physics. The authors demonstrate giant THz Hall conductivity with small imaginary components at room temperature, and claim the dissipation-less nature of the anomalous Hall effect in this non-collinear antiferromagnet. Although overall story is clear and is mostly supported by the experimental data shown in Figs. 1 and 2, I would like to ask the authors to consider the comments/questions appended below.

1. For general readers, please illustrate how the antiferromagnetic spin dynamics at THz frequency

lead to (or related to) the giant Hall conductivity in Mn₃Sn.

Also please compare the THz Hall angle presented in this manuscript to those in graphene and topological insulators etc. measured by the similar techniques.

2. Please provide information on the sample characterization and on the definition of “high quality”. It would also be helpful to add the location of Weyl points (how far from the E_F) for the studied films in the main text.

3. The discussion is given solely on the low-temperature spin-glass phase, which is rather separated from the main story of “the giant THz Hall effect”. Please revise the main text by emphasizing the main story, and the scientific/technical importance of THz Hall effects in antiferromagnets, by adding more details on the electronic structures of Mn₃Sn.

4. In Fig. 2(c), the Hall angle seems to stay at a finite value when we extrapolate the film thickness down to zero. Please elaborate this point.

5. In Fig. 2(d), the imaginary part for $x = 0.04/\text{Si}$ goes to negative, even considering the error bar. Is this reasonable?

6. The oscillations in Figs. 3(c) and 3(d) are due to the experimental noise?

7. Please revise Fig. 3(f). It is difficult to catch the spin structures illustrated.

Minor:

Line 204: “increases slightly below 50 K”.

It seems that the increase is discernible below 150 K.

In conclusion, I would not recommend this manuscript to be published in Nature Communications in the current form.

Reply to Reviewer #1

The authors present an experimental follow-up work on their earlier ground-breaking observation (Ref. 7) of the AHE in a non-collinear antiferromagnet Mn₃Sn. They now extend the AHE measurements from DC to the THz frequency range. The employed THz method and the presented data appear to be valid. In the present form of the manuscript, however, the authors did not evidence or explain in a transparent way a new unexpected phenomenology of the AHE in Mn₃Sn or help elucidating physical origins of the previously observed phenomenology. More specifically, I have the following comments:

We would like to thank Reviewer #1 for evaluating our manuscript highly and giving us constructive feedback and comments. We have carefully read and responded to all the comments. The one-to-one replies are as follows:

1) Terminology: In their pioneering work from 2015 (Ref. 7), they call the DC AHE in single crystal Mn₃Sn a large effect. The observed DC AHE reached 30 Ohm⁻¹cm⁻¹ at room-T and 100 Ohm⁻¹cm⁻¹ at 100 K. In the present manuscript the values reach 20 Ohm⁻¹cm⁻¹. Also in comparison, in ferromagnetic Fe, AHE is around 1000 Ohm⁻¹cm⁻¹. The authors should therefore justify the use of the term “giant/gigantic” AHE in the present manuscript or use a more appropriate terminology.

We believe that the largeness of the AHE in Mn₃Sn far exceeding typical antiferromagnets and comparable to ferromagnets is an important point in this work and should be emphasized. As Reviewer #1 points out, however, the word “giant/gigantic” might not be appropriate from the viewpoint of the fact that the even larger value in a single crystal was called “large” in the previous paper. Therefore, according to the Reviewer’s comment, we call “large” AHE instead of “giant/gigantic” in the whole revised manuscript. We appreciate Reviewer #1 for the valuable comment.

2) The authors explore the frequency range of 2-27 meV which is well below their experimental value of the Drude scattering rate at room-T of 60meV (tau=10fs). So their AC experiment is deep in the Drude peak, i.e., still close to the DC limit. Under these conditions, the expected result is that the THz AHE would be similar to the DC AHE which at room-T indeed seems to be the case. The authors should note this in the manuscript or, if they feel that the observation was unexpected, they should explain it more explicitly.

It is a very important comment related to the value of our manuscript and we would like to explain it in detail.

As suggested by Reviewer #1, the THz longitudinal conductivity $\sigma_{xx}(\omega)$ is indeed close to the DC limit because the energy scale of 2-27 meV is smaller than the Drude scattering rate. In the same way, one might expect that the THz anomalous Hall conductivity $\sigma_{xy}(\omega)$ would be also in the DC limit due to the very same reason. However, we would have to say that it is not correct, because *the mechanism of σ_{xy} is in principle totally different from that of σ_{xx}* , especially for the scattering mechanisms. σ_{xx} is dissipative in the DC limit and is usually described with the Drude model where any kinds of momentum scattering process can be involved. On the other hand, σ_{xy} is dissipationless in the DC limit and comes from the Berry curvature in momentum space if it is intrinsic and it therefore occurs without any scattering. σ_{xy} can also originate from the impurity-induced scattering if extrinsic. There is no direct connection between $\sigma_{xx}(\omega)$ and $\sigma_{xy}(\omega)$.

In particular, in addition to the Weyl semimetallic bands, Mn_3Sn also has other metallic bands across the Fermi energy as studied in Ref. 34 and others. One can naturally expect that the longitudinal conductivity $\sigma_{xx}(\omega)$ and the relevant scattering rate are mainly dominated by the carriers in the metallic bands around the Fermi level. The anomalous Hall conductivity $\sigma_{xy}(\omega)$ is, on the other hand, relevant to the integrated Berry curvature at the occupied states over the entire Brillouin zone. Therefore, it is challenging to predict how σ_{xy} depends on frequency even if the scattering rate in σ_{xx} is known.

Let us also explain the previous studies of the THz AHE. In the study of the THz AHE in a ferrimagnetic alloy $\text{Gd}_x(\text{FeCo})_{1-x}$ in Ref. 27, the AHE was assumed to be described with the Drude model under external magnetic field as in the same way with the normal Hall effect, and then the external magnetic field was replaced with a phenomenological parameter $\lambda^* = eB$ to describe the Hall effect without magnetic field. Therefore, Ref. 27 shows that the THz AHE is in the DC limit in a very similar way to the longitudinal conductivity. Note that such a frequency dependence of the AHE is just a phenomenological assumption and not motivated theoretically. This study was focused not on discussing the frequency dependence of the AHE but instead on investigating ultrafast dynamics of the ferrimagnet by utilizing the THz AHE as a time-resolved probe. Therefore, the phenomenological approach in Ref. 27 captures the essential physics. In our manuscript, on the other hand, the aim of the research is “to investigate the frequency dependence of the AHE in THz range” for future fast spintronics based on antiferromagnet. Furthermore, the large AHE in Mn_3Sn is not scaled with its vanishingly-small net magnetization and therefore more microscopic picture is required to understand. For this purpose, the previous phenomenological assumption is not justified.

In another experiment in Ref. 22, spectral features of the AHE in the THz frequency range have been investigated for ferromagnet SrRuO_3 . While σ_{xx} of this material in THz frequency was similar to the DC value, the THz σ_{xy} was several times larger than DC one. The significant deviation of σ_{xy}

between THz and DC was explained by possible existence of lower-energy excitations within the interpretation of the intrinsic mechanism. Although $\sigma_{xx}(\omega)$ also showed a deviation from the Drude model, it was not large and not related to the significant difference of $\sigma_{xy}(\omega)$ between THz and DC. It is a good example of the fact that the frequency dependence of the AHE is not straightforward and difficult to predict from the DC conductivity.

As long as one considers the intrinsic AHE, the frequency dependence of σ_{xy} would be described by the interband transition across the band (anti)crossing point, and therefore whether $\sigma_{xy}(\omega)$ is in the DC limit or not is determined by the electron band structure. In our Mn_3Sn film, the Weyl nodes are a few meV above the Fermi energy, i.e., in the THz frequency range. From this viewpoint, one could expect that substantial frequency dependence due to the interband transitions might appear in THz energy scale, as studied theoretically in Refs. 29-33. As a result, we found that the THz AHE coincides with the DC AHE in Mn_3Sn and that sharp spectral features were absent, which might be ascribed to the effect of thermal energy. Again, we would like to stress that it is not directly related to the scattering rate measured in σ_{xx} .

According to the comments by Reviewer #1, we added the explanation above in the revised manuscript to clearly describe that, although the THz longitudinal conductivity is in the DC limit, it does not mean that the THz AHE is also in the DC limit because the relevant scattering mechanisms are different. We thank Reviewer #1 for the valuable comment to improve explanation of our manuscript.

3) As the authors mention in the paper, performing AHE measurements at THz is not novel on its own as this was done earlier in other magnetic systems. However, it is not a straightforward technique so the authors certainly do deserve a credit for their work here. On the other hand, because this is an elaborate technique, the reader might wonder how relevant are the remarks made by the authors on the relevance of their work for future high frequency device applications. The authors should elaborate in more detail on this potential relevance.

We thank Reviewer #1 for their evaluating the value of our work. For application of data processing in spintronic devices, how one can read the spin information in antiferromagnets has been a problem to be resolved. We believe that ultrafast readout of THz AHE in antiferromagnets in this manuscript is a substantial importance apart from earlier observation of THz AHE in ferromagnets.

As Reviewer #1 points out, the elaborate scheme of THz polarization-resolved spectroscopy is not directly used for application in THz electronics and spintronics, since the far-field light irradiation seems not to be suited because of the large spot size due to the sub-mm wavelength. For a practical use, THz anomalous Hall conductivity must be measured in much smaller region. Very recently, light-induced THz AHE in a very tiny graphene flake has been recently detected via contact electrodes

combining with optical pulses and THz current generation on chip (J. W. McIver *et al.*, arXiv:1811.03522, online published in Nature Phys.). We can expect that the large THz AHE in Mn_3Sn would be measured in a similar way on integrated devices.

On the other hand, our polarization-resolved THz far-field spectroscopy based on photonics technology is more directly related with ultrafast time-resolved spectroscopy to reveal nonequilibrium dynamics of Weyl antiferromagnets for fundamental interests. The all-optical approach in a noncontact way is a powerful tool to reveal intrinsic properties of materials. We believe that the demonstration of THz AHE in this manuscript will facilitate pump-probe studies of Weyl semimetals in THz photonics as well as ultrafast manipulation and readout of the spins in THz electronics.

According to the comment by Reviewer #1, we revised our manuscript to clearly state how the present spectroscopy of THz AHE can be related to such a high-frequency device application with electrodes. It is also related to the comments #1 and #3 of Reviewer #3.

4) The seemingly surprising result is the drop of the THz AHE below 250K. The authors associate this observation with a transition to a helical state. However, they do not provide an additional experimental verification for this interpretation. Fig. 3c shows an upturn of the AHE at low frequencies which seems to extrapolate to a large value in the DC limit. This seems inconsistent with the scenario of the phase transition into the helical state. It would be very helpful to see the AHE measured in these samples at lower temperatures as well. In Ref. 7, the DC AHE in single crystal Mn_3Sn does not show a drop at lower temperatures.

We carefully read this comment and speculate that different aspects of the phase transitions in this material might have confused Reviewer #1.

First of all, we explain the validity of our interpretation of the reduction of the THz AHE below 250 K observed in this work. The reduction of the AHE was also observed in DC in the similar film sample in Ref. 35, which was also accompanied with the reduction of magnetization. Such a phase transition of magnetism in Mn_3Sn at around 250 K has been well studied by the neutron scattering and magnetic moment measurements in Refs. 43-48 (the reference numbers are those in the revised manuscript), which have revealed the formation of helical (or spiral) spin ordering along the c -axis. The results of the thin film in Ref. 35 were fully consistent with this picture of the phase transition. In this work, we have also performed the DC anomalous Hall measurement for the samples fabricated in the same method and confirmed that the data were consistent. Therefore, we reasonably interpreted that the reduction of the THz AHE below 250 K in this work is also due to the spin-reorientation phase transition to the helical state.

As indicated by Reviewer #1, the single crystal Mn_3Sn used in Ref. 7 did not show the transition to the helical state below around 250 K. It was because the sample used in Ref. 7 has been made

intentionally to *prevent* this transition to the helical state. It was realized by not annealing below 800 degree Celsius as described in the Methods section in Ref. 7. This process stabilizes the inverse triangular spin configuration at least down to 50 K for the bulk single crystal, and therefore the temperature dependence in Ref. 7 is different from the film used in the present work.

Then, let us consider another transition at much lower temperature. Even for the bulk single crystal in Ref. 7, another transition occurs below 50 K. Such a transition has been also studied in the neutron scattering in Refs. 43, 46-48 of the revised manuscript, which have revealed that the spins cant slightly towards the *c*-axis to form the ferromagnetic spin-glass state. In our present manuscript, at the low temperature we observed the upturn of the low-frequency AHE in Fig. 3(c) as raised by Reviewer #1. Since the upturn is most pronounced at the lowest temperature where the spins cant along the *c*-axis, the upturn is apparently related to the ferromagnetic spin-glass state, not to the helical state. Therefore, although Reviewer #1 raised a concern about inconsistency for the transition to helical phase, we believe that the interpretation of the phase transition is fully consistent between our work and the previous studies.

As to the transition to the ferromagnetic spin-glass state, we observed the upturn in the THz AHE spectrum. For comparison, according to the suggestion by Reviewer #1, we added the data of the DC AHE for the same sample on the left axis ($\omega = 0$) in Figs. 3(a) and 3(c) in the revised manuscript. The THz AHE deviated from the DC AHE at the low temperatures. Such a frequency dependence of AHE could suggest existence of another lower-energy excitation in this ferromagnetic spin-glass phase. However, our THz AHE measurement was performed without external magnetic field on cooling the sample. Since our high-resolution THz polarimetry setup has not been implemented with a superconducting magnet, the sample had been magnetized at room temperature in advance at a different place, and then it was slowly cooled down in our THz polarimetry setup at zero magnetic field. Because of the absence of external field on cooling process, the spin-reorientation phase transition is inevitably accompanied with formation of magnetic domains. Therefore, our THz spectroscopy corresponds to the measurement of an effective response function in a spatially-inhomogeneous system at the low temperature. To treat the response function of such a system, the effective medium theory is known to be valid. Previously THz spectra for spatially-inhomogeneous systems have been discussed with the effective medium theory, in which peculiar spectral features could appear (See *e.g.*, P. Kužel and N. Němec, J. Phys. D: Appl. Phys. 47, 374005 (2014)). Our experimental observation for the upturn of THz AHE may also be induced by such an extrinsic spatial inhomogeneity.

If our THz polarimetry is realized under magnetic field, we expect that low-energy response at the low-temperature phase would be clarified in detail with macroscopic magnetic domain. Please note that, however, the upturn of σ_{xy} in Fig. 3(c) is a very tiny signal as small as a few $\Omega^{-1}\text{cm}^{-1}$, which corresponds to the rotation angle of ~ 500 μrad . To clearly observe and discuss such a small spectral

feature, the polarization resolution has to be much better (~ 50 μrad). Such a high resolution of the polarization rotation was realized in this work with our elaborated THz polarimetry scheme. In our present condition, unfortunately, such a precise-resolution spectroscopy was not possible under magnetic field. In the present manuscript, we focus on the realization of the ultrafast readout of the THz AHE in the Weyl antiferromagnet at room temperature for practical application. It should be also an important topic on its own as Reviewer #2 suggested. Reviewer #3 also recommended that our discussion should be more focused on the room temperature rather than the low-temperature phase. We believe that the further investigation of the low-temperature phase by implementing precise-resolution THz polarimetry under magnetic field is beyond the scope of this manuscript and to be investigated in future work.

In the revised manuscript, we modified the explanation of the reduction of the AHE around 250 K with adding the references and the data for DC AHE in Figs. 3(a) and 3(c).

Reply to Reviewer #2

The authors report on very interesting work probing a terahertz anomalous hall response in Weyl antiferromagnetic systems. The all-optical approach is powerful here and as the authors state, opens up new possibilities for probing the non-equilibrium response. The clear evidence for dissipationless-like transport is also quite important. I only have a few minor suggestions which I think could improve the paper.

We would like to thank Reviewer #2 for highly esteeming our manuscript and also recommending publication only with small revisions. We have carefully read and responded to all the suggestion. The specific replies are as follows:

1. The authors write in the abstract "Observation of the THz AHE at room temperature demonstrates the ultrafast readout for the antiferromagnetic spintronics using Mn₃Sn, paving the path for ultrafast control of magnetism in Weyl antiferromagnets." My view is that the paper stands on its own without this last comment in the abstract which to me is a slight oversell. The approach does enable connection to pump-probe studies as noted above but does not address the means by which control of this system would be enabled. I think it's sufficient to describe this in the final discussion session as is currently done.

We thank Reviewer #2 for evaluating the value of our results highly. As suggested by Reviewer #2,

we modified the abstract in the revised manuscript to properly describe the achievement of our work at this stage as following:

“Observation of the THz AHE at room temperature demonstrates the ultrafast readout for the antiferromagnetic spintronics using Mn_3Sn and will also open new avenue for studying nonequilibrium dynamics in Weyl antiferromagnets.”

2. In ref 35 by the same authors, they see a similar drop in scattering time around 250K in the unmagnetized state. I would like to understand the relation here between these two measurements. Are these expected to give similar results?

We interpreted that Ref. 35 in this comment is a typo for Ref. 36.

To compare the results of two measurements, we plotted the temperature dependence of the scattering rate $1/2\pi\tau$ below. The gray circles are the data in the inset of Fig. 1(f) in the revised manuscript and the red triangles are the data in Ref. 36.

The results are quite similar, and it is natural because these thin-film samples were made in the similar methods reported in Ref. 35. Therefore, in the main text we described the present work “well reproduces the previous THz-TDS measurement” with respect to the temperature dependence of the scattering rate.

3. I suggest that Figure 1e include more information in the figure caption itself rather than just referring the reader to the text. Right now this is not easily understandable.

According to the suggestions by Reviewer #2, we added more detailed information in the figure caption of Fig. 2(a) in the revised manuscript as following;

“(a) Frequency dependence of the precision of the polarization rotation angle in this measurement evaluated by the standard deviation of the polarization rotation angle. The precision can be more improved with using a larger number (#) of data sets. For example, the precision for 20 data sets (#20) can be as small as several tens of μrad between 0.5 and 2.0 THz (See details in text and Methods)”.

4. The phrase "We also investigated the temperature dependence of the THz AHE for $\text{Mn}_{3+x}\text{Sn}_{1-x}$ films ($x=0.02$) on a SiO_2 substrate at zero-field cooling after the magnetization at 300 K" needs rewording I think.

We thank Reviewer #2 for giving the comment to improve the text. According to the comment, in the revised manuscript we changed the sentence as following;

“We also measured the temperature dependence of the THz AHE for $\text{Mn}_{3+x}\text{Sn}_{1-x}$ films ($x=0.02$) on a SiO_2 substrate. The sample was magnetized at 300 K in advance and cooled down under zero magnetic field.”

Reply to Reviewer #3

This manuscript by T. Matsuda et al. reports polarization-resolved THz spectroscopy on a Weyl semimetal candidate Mn_3Sn , which is of current interest in the field of condensed matter physics. The authors demonstrate giant THz Hall conductivity with small imaginary components at room temperature, and claim the dissipation-less nature of the anomalous Hall effect in this non-collinear antiferromagnet. Although overall story is clear and is mostly supported by the experimental data shown in Figs. 1 and 2, I would like to ask the authors to consider the comments/questions appended below.

We would like to thank Reviewer #3 for highly esteeming our manuscript and giving us constructive feedbacks and comments. We have carefully read and responded to all the suggestion. Our replies are as follows:

1. For general readers, please illustrate how the antiferromagnetic spin dynamics at THz frequency lead to (or related to) the giant Hall conductivity in Mn_3Sn . Also please compare the THz Hall angle presented in this manuscript to those in graphene and topological insulators etc. measured by the

similar techniques.

Thank you for the valuable comment to improve our manuscript. It should be noted more clearly that the large AHE in Mn_3Sn is not directly related to the vanishingly-small antiferromagnetic spin moment and it is attributed to the cluster octupole moment in the Kagome bilayer. The spin motions in the Kagome bilayer are characterized by six precessional eigenmodes, one of which corresponds to collective precession of all the spins in-phase within the ab plane. When a strong in-plane magnetic field is applied, all of the spins change their directions simultaneously according to the field, which corresponds to the damping of the acoustic collective precessional mode. Such a directional change of all the spins corresponds to that of the cluster octupole moment. This motion would be expected to occur as fast as in picosecond timescale because of the exchange interaction in the antiferromagnetic metal. Therefore, if intense THz magnetic field could flip the cluster octupole moment, it would result in the sign reversal of the large AHE. In the conclusion of the revised manuscript we added this explanation as a perspective.

Reviewer #3 also requests us to compare the THz Hall angle ($\sim 4\text{-}7$ mrad) observed in this manuscript to those in graphene and topological insulators measured by the similar techniques. Please note that the Hall conductivity requires the breaking of time-reversal symmetry, and that the Hall conductivity in graphene and topological insulators are therefore induced by applying an external magnetic field for the symmetry breaking (“normal Hall effect”). On the other hand, here we investigate the “anomalous Hall effect” that occurs without external magnetic field due to the broken symmetry inherent to the material itself. These two phenomena are essentially different. The “normal” Hall angle is dependent on both the strength of applied magnetic field and the carrier density (or sample quality). Effect of magnetic field is not even monotonic because the relation between the frequency of light and the cyclotron frequency is also closely related. Therefore, we would have to say that the direct comparison of the Hall angles between the anomalous Hall effect in Mn_3Sn and the normal Hall effect in these other systems are not so meaningful.

But for one’s information, a giant THz Faraday rotation as large as 100 mrad at $\hbar\omega \sim 10$ meV has been reported in graphene with the Fermi energy ~ 340 meV by applying 7-T magnetic field (I. Crassee *et al.*, Nature Phys. 7, 48 (2011)). Another graphene sample with smaller Fermi energy ~ 60 meV shows only 5-mrad rotation for $\hbar\omega \sim 4$ meV at 7 T in Ref. 40, in which the Faraday rotation is in the quantum regime. Such a large polarization rotation and quantization have been also reported in topological insulators.

It should be also noted that ~ 2 mrad rotation by the THz “anomalous” Hall effect has been reported in a 10-nm thin film of a ferromagnet SrRuO_3 at 10 K in Ref. 22. Compared to this, the rotation angle of 4-7 mrad in the *antiferromagnet* in our manuscript is a bit larger, but the value also could depend on the sample thickness. Therefore, we would not add the comparison of the values of the Hall angles.

2. Please provide information on the sample characterization and on the definition of “high quality”. It would also be helpful to add the location of Weyl points (how far from the E_F) for the studied films in the main text.

To provide further information of the characterization of our samples, in the revised manuscript, we added our XRD (Fig. 1(c)) and DC anomalous Hall conductivity (Fig. 3(a) and 3(c)) in $\text{Mn}_{3+x}\text{Sn}_{1-x}$ ($x=0.02$) on a SiO_2 substrate. Our samples were fabricated according to the method in Ref. 35. Here the value of the AHE at room temperature is comparable to the large value reported for single crystals in Ref. 7. In this sense, we call it “high quality” in the original manuscript. But it might be not clear for readers. Therefore, we modified description of the sample in the main text.

Previously the locations of the Weyl points have been investigated in Ref. 34 with the first-principle calculation and ARPES measurement. The Weyl points were assigned to be at 8 meV above the Fermi energy E_F for $\text{Mn}_{3+x}\text{Sn}_{1-x}$ single crystal with $x=0.03$. In this manuscript, we mainly used a film sample with $x=0.02$. Since the Weyl-point energy measured from E_F increases according to the increase of Mn composition x , the Weyl points in our film samples are also expected to be similar. For the thin films, however, it is very challenging to identify the location of the Weyl points because the evaluation of the absolute value of the anomalous Nernst effect or ARPES measurement is difficult at the present stage. In our measurement, we investigated the THz AHE for the samples with $x=0.00, 0.02, \text{ and } 0.08$, and then found that the results are very similar for every sample, suggesting that fine energy structure of the Weyl points are obscured due to the thermal broadening in type-II dispersion.

In the revised manuscript, we added the explanation for the location of the Weyl points in the sample characterization part in the Methods section.

3. The discussion is given solely on the low-temperature spin-glass phase, which is rather separated from the main story of “the giant THz Hall effect”. Please revise the main text by emphasizing the main story, and the scientific/technical importance of THz Hall effects in antiferromagnets, by adding more details on the electronic structures of Mn_3Sn .

The previous version of our manuscript was divided into the four sections as “Sample”, “THz spectroscopy”, “Results”, and “Discussion”. As pointed by Reviewer #3, however, the main story of the large THz AHE at room temperature was discussed in the “Results” section, while the “Discussion” section was focused only on the low-temperature glass phase experiment. Such a categorization was indeed not appropriate for the main focus of this paper. Therefore, according to the comment by Reviewer #3, we modified the Results and Discussion sections in the revised manuscript and created new sections as “Large THz anomalous Hall effect at room-temperature” and “Temperature

dependence of THz anomalous Hall effect". We believe that the revised text properly describes the main story of this manuscript. We would like to thank Reviewer #3 for improving our manuscript.

For the scientific and technical importance of the THz AHE in antiferromagnets, we revised the conclusion part of this manuscript to clearly state how our approach is important for ultrafast readout of spin information in antiferromagnets and also for study of nonequilibrium dynamics and control of Weyl semimetals. This revision is also related to the comment #3 by Reviewer #1.

We described that **"our all-optical approach in the noncontact way with picosecond time resolution can be extended to pump-probe measurements for the study of nonequilibrium dynamics"** and that **"our demonstration of the THz AHE will lead to such a readout of the spin information on integrated devices"**.

4. In Fig. 2(c), the Hall angle seems to stay at a finite value when we extrapolate the film thickness down to zero. Please elaborate this point.

The relation between the rotation angle θ and the film thickness d was shown in Eq. (1) of the main text as $\theta = \sigma_{xy}Z_0d/(1 + n_s + \sigma_{xx}Z_0d)$. Here we also plot the calculated curve of Eq. (1) using the experimental DC value $\sigma_{xx} = 3800 \Omega^{-1} \text{ cm}^{-1}$, $\sigma_{xy} = 19 \Omega^{-1} \text{ cm}^{-1}$ for 200-nm-thick film.

In the small thickness regime, the rotation angle θ can be approximated as $\theta \sim \sigma_{xy}Z_0d/(1 + n_s)$ and therefore $\theta \propto d$. On the other hand, in the large thickness regime, θ is saturated to be $\theta \sim \sigma_{xy}/\sigma_{xx}$. In our experiment, the 200- and 400-nm samples are in the saturated regime, while the 50-nm sample is in the crossover regime.

The calculated curve of Eq. (1) assumes that the DC values of σ_{xx} and σ_{xy} are independent of film thickness. But in the experiment the σ_{xx} and σ_{xy} were found to be slightly different for each film thickness. Therefore, in the original manuscript we have plotted the calculated results as the white circles with using the real values of σ_{xx} and σ_{xy} for each thickness film. However, the difference

between the white circles and the curve in the figure above is small enough. Therefore, for clarity, we replaced the white circles by the solid curve in the revised manuscript to clearly show the thickness dependence of the rotation angle. We have now added a brief explanation of the thickness dependence to the manuscript.

5. In Fig. 2(d), the imaginary part for $x = 0.04/\text{Si}$ goes to negative, even considering the error bar. Is this reasonable?

Since the imaginary part of $\sigma_{xy}(\omega)$ corresponds to the difference between the absorption of clockwise and counterclockwise circularly polarized light, in principle the value can be both positive and negative. However, considering the previous theory for a type-II Weyl semimetal in Ref. 32, the low-frequency light absorption around the Weyl nodes occurs preferentially only for one circular polarization direction. In this sense, the data going to negative in a very limited frequency range in Fig. 2(d) (corresponding to Fig. 2(f) in the revised manuscript) seems not to be reasonable.

In this work, the error bars were evaluated only with considering statistical errors in our experimental data accumulation. But another systematic error could also appear in experiment, which could originate from artifacts such as long-term laser instability. Here the negative value of $\text{Im } \sigma_{xy} \sim -3 \Omega^{-1}\text{cm}^{-1}$ at around 18 meV (~ 4.5 THz) corresponds to the rotation angle of as small as ~ 500 μrad . Note that this frequency is beyond the frequency window of our high-precision THz polarimetry (0.5-2.0 THz) shown in Fig. 1(e) (corresponding to Fig. 2(b) in the revised manuscript). Such an accurate measurement of the rotation angle in this high frequency has been still challenging. We have performed 4-hour accumulation of the broadband measurement, which is inevitably sensitive to

fluctuation of the laser output.

The broadband $\sigma_{xy}(\omega)$ spectrum in Fig. 2(d) was measured with a sample on a Si substrate with $x=0.08$. For comparison, we also performed the same measurement for another sample with $x=0.00$ in figure above. In this sample, the negative $\sigma_{xy}(\omega)$ was hardly seen. Therefore, we interpreted the data points of the negative imaginary part for $x=0.08$ as some artifacts. Note that the overall spectral feature in the imaginary part which gradually increases toward the higher frequency side is consistent for both samples, indicating intrinsic interband transition across the nodes.

6. The oscillations in Figs. 3(c) and 3(d) are due to the experimental noise?

As commented by Reviewer #3, the oscillations can be seen in Figs. 3(c) and 3(d), and also in Figs. 3(a) and 3(b). Such an oscillation could be observed in THz time-domain spectroscopy due to interference of THz pulses measured in time domain. We have carefully investigated our data and found that such oscillations seem to appear when we have used our cryostat. For example, the oscillation is not noticeable in Figs. 2(a) and 2(b) (corresponding to Fig. 2(c) and 2(d) in the revised manuscript) for which the cryostat was not used at room temperature. We expect that some pre-pulse in the femtosecond laser may generate another THz pulse, which could be reflected from the cryostat window and might be detected in the time-domain spectroscopy together with the main THz pulse. To completely eliminate it was still difficult. We believe that it is small enough for discussion in our manuscript.

In the revised manuscript, we mentioned the origin of the oscillation in Figs. 3(a)-3(d) in the Methods section.

7. Please revise Fig. 3(f). It is difficult to catch the spin structures illustrated.

According to the comment by Reviewer #3, we modified Fig. 3(f) (corresponding to Fig. 3(e) in the revised manuscript) to clearly show the spin structure.

Minor: Line 204: “increases slightly below 50 K”. It seems that the increase is discernible below 150 K.

The phase transition temperature varies with Mn composition or thin film quality. As Reviewer #3 points out, Fig. 3(e) shows that the real part of the Hall conductivity spectrum seems to start increasing slightly from 150 K. In the revised manuscript, we rewrote “... **increases slightly below 150 K**”.

REVIEWERS' COMMENTS:

Reviewer #1 (Remarks to the Author):

After reviewing the responses of the authors to the previous comments I find the work now suitable for publication.

Reviewer #2 (Remarks to the Author):

The authors have done a careful job of replying to all comments. I recommend publication at this stage.

Reviewer #3 (Remarks to the Author):

The authors' replies to the comments/questions raised by the reviewers seem to be reasonable. Therefore, now I would like to recommend their revised manuscript to be published in Nature Communications.

Point-by-point responses:

Reviewer #1 (Remarks to the Author):

After reviewing the responses of the authors to the previous comments I find the work now suitable for publication.

We deeply appreciate Reviewer #1 for his/her previous comments and criticisms which improved our manuscript and also for the present comment recommending publication of our manuscript without any changes.

Reviewer #2 (Remarks to the Author):

The authors have done a careful job of replying to all comments. I recommend publication at this stage.

We deeply appreciate Reviewer #2 for highly evaluating the experimental result of our manuscript and also for the present comment recommending publication without any changes.

Reviewer #3 (Remarks to the Author):

The authors' replies to the comments/questions raised by the reviewers seem to be reasonable. Therefore, now I would like to recommend their revised manuscript to be published in Nature Communications.

We deeply appreciate Reviewer #3 for his/her previous comments and constructive feedbacks which was very helpful for us, and also for the present comment recommending publication of our manuscript without any changes.